# What Are the Most Effective Factors in Determining Future Exacerbations, Morbidity Weight, and Mortality in Patients with COPD Attack?

**DOI:** 10.3390/medicina58020163

**Published:** 2022-01-21

**Authors:** Çağla Koç, Füsun Şahin

**Affiliations:** 1Erbaa State Hospital, Ministry of Health, 60500 Tokat, Turkey; caglakoc23@gmail.com; 2Department of Chest Diseases, Yedikule Chest Diseases and Thoracic Surgery Training and Research Hospital, University of Health Sciences, 34760 İstanbul, Turkey

**Keywords:** COPD, biomarker, exacerbation, morbidity, mortality

## Abstract

Background and Objectives: This study aimed to investigate the important factors that affect COPD prognosis. Materials and Methods: We included 160 hospitalized patients with COPD exacerbation in the study. The hemoglobin (HB), hematocrit (HCT), leukocytes, red cell distribution width (RDW), mean platelet volume, platelet distribution width, plateletcrits, platelets, neutrophil/lymphocyte ratio, platelet/lymphocyte ratio, eosinophils, uric acid, albumin, C-reactive protein (CRP), procalcitonin, arterial blood gases (PO2 and PCO2), pulmonary function test (FEV1 and FVC), echocardiography (ejection fraction-EF), Global Initiative for Chronic Obstructive Lung Disease (GOLD) stage, Modified Medical Research Council (mMRC) and Borg scales, Charlson comorbidity index, body mass index (BMI), and the length of hospital stay were examined on the first day of hospitalization. Admission to the hospital with a new attack, hospitalization in the intensive care unit (ICU), and mortality during the six months after discharge were evaluated. Results: High CRP and procalcitonin levels were observed in the group with a long hospital stay. In the mortality group, the HB, HCT, BMI, and PO2 values were significantly lower than in the group without mortality, while the age and GOLD stage were higher. The age, Borg and mMRC scores, number of exacerbations experienced in the previous year, RDW, eosinophil count, and PCO2 were significantly higher in the ICU group than that without an ICU stay. The HCT and EF values were lower in the ICU group than that without an ICU stay. The FEV1 and FVC values were significantly lower in the follow-up attack group than those without a follow-up attack. The duration of COPD and the number of attacks that were experienced in the previous year were high. Conclusion: Scoring combining selected biomarkers and other factors is a strong determinant of the prognosis.

## 1. Introduction

Chronic obstructive pulmonary disease (COPD) is a common, preventable, and treatable disease that is characterized by persistent respiratory symptoms and airflow limitations due to airway and/or alveolar abnormalities that are usually caused by significant exposure to noxious particles or gases [1]. An exacerbation of COPD is defined as an acute worsening of respiratory symptoms that result in additional therapy. Exacerbations affect the state of health, hospitalization, and also the course of the disease negatively [1]. The most important predictor of frequent exacerbations is previous exacerbations [2]. An aggravation of airway obstruction causes an increase in the exacerbation frequency, hospitalization, and death risk [2,3,4]. Hospitalization for COPD exacerbation is associated with a poor prognosis and an increased risk of death [5]. According to the GOLD (Global Initiative for Chronic Obstructive Lung Disease) staging, in which airflow limitation is graded according to the first second of forced expiration (FEV1), the risk of exacerbation increases as the stage increases, but FEV1 alone does not predict exacerbation or mortality [2,3]. Severe exacerbations that require hospitalization are especially associated with high mortality due to COPD. Considering that acute exacerbation of COPD plays an important role in the prognosis, effective markers are sought to determine the acute attack frequency, length of hospitalization, severity of morbidity, and mortality that can be easily obtained at the time of admission of patients that are presenting with COPD exacerbation [6].

Until recently, COPD was defined as a progressive lung disease that was characterized by airflow restriction, but recently it has been shown that airflow restriction is associated with an abnormal inflammatory response and COPD is not limited to the lungs but also has systemic effects [7]. Especially during the exacerbation period of COPD, markers showing systemic inflammation in the blood increase [1]. Studies have shown the importance of systemic inflammation in the pathogenesis of COPD and some inflammatory biomarkers such as the neutrophil/lymphocyte ratio (NLR), uric acid, CRP (C-reactive protein), and procalcitonin (PCT), which are indicators of inflammation that are associated with the severity of the disease during the exacerbation period. In our study, we aimed to investigate the efficacy of inflammatory biomarkers and other factors [arterial blood gas, PFT (pulmonary function test), EF (ejection fraction), PAP (pulmonary arterial pressure) values, body mass index (BMI), Charlson Comorbidity Index, mMRC (Modified Medical Research Council), and the Borg scale] in predicting the frequency of acute attacks, duration of hospitalization, morbidity, and mortality in patients that were hospitalized due to COPD exacerbation.

## 2. Materials and Methods

### 2.1. Patient Population

Our study included a prospective cohort; 160 patients who were hospitalized due to COPD exacerbation at the University of Health Sciences/Yedikule Chest Diseases and Thoracic Surgery Training and Research Hospital between 1 July 2018 and 1 March 2019, and who met the inclusion criteria were evaluated. The hospitalization date, demographic information (age, gender), day of hospitalization, hemogram, CRP, PCT, uric acid, total protein, albumin, urea, creatinine, arterial blood gas, PFT (pulmonary function test), EF (ejection fraction), PAP (pulmonary arterial pressure) values, body mass index (BMI), Charlson comorbidity index, mMRC (Modified Medical Research Council), and Borg scores of all the patients were recorded. COPD duration, smoking habits, use of BPAP (bilevel positive airway pressure) and an oxygen concentrator, exacerbation in the previous year and hospital, and ICU (intensive care unit) hospitalization history were investigated and recorded. The patients were re-evaluated in the first six months after their discharge from the hospital, and information about exacerbations, hospitalizations, and deaths during this period was recorded on a monthly basis. The inclusion criteria were as follows: patients that were aged at least one year old and were diagnosed with COPD in the exacerbation period. The exclusion criteria were as follows: COPD patients that were in a stable period; cancer patients; patients with kidney or liver failure; patients with a hematological disease; patients with active bleeding or who had a blood transfusion in the last three months; patients with a pulmonary embolism, acute myocardial infarction, or cerebrovascular disease in the last month; and patients with a myeloproliferative or autoimmune disease history.

Our study was conducted in accordance with the Declaration of Helsinki, with the approval of the Scientific Committee of the University of Health Sciences/Yedikule Chest Diseases and Thoracic Surgery Training and Research Hospital and the Ethics Committee of the University of Health Sciences/İstanbul Training and Research Hospital (approval date: 22 June 2018, decision no.: 2018/1309). Informed written consent for study participation was obtained from each subject.

### 2.2. Assessment of Symptoms and Comorbidities

The diagnosis of COPD was made according to the post-bronchodilator FEV1 (first second of forced expiration)/FVC (forced vital capacity) <70% criteria by clinical evaluation and spirometry according to the GOLD COPD guideline. The exacerbation period was determined according to the presence of acute worsening of respiratory symptoms requiring additional treatment and the Anthonisen criteria. The patients with COPD who were hospitalized during the exacerbation period were included in the study. The classification that was determined by Anthonisen in 1987 is often used to determine the COPD exacerbation severity [8]. Exacerbations are classified as Group I, Group II, and Group III. The presence of three main criteria: dyspnea, sputum purulence, and increased sputum amount is classified as Group I (severe; requiring emergency admission or hospitalization), the presence of two criteria is classified as Group II (moderate; requiring treatment with short-acting beta2 agonists as well as antibiotics and/or oral corticosteroids) and patients are classed as Group III (mild exacerbation; requiring treatment with only short-acting beta2 agonists) if there is only one criterion that is met and there is at least one of the following features: a recent upper respiratory tract infection, fever, wheezing, a cough, an increased respiratory rate, or a heightened pulse rate [8]. “Frequent exacerbation” is defined as having two or more moderate exacerbations or exacerbations requiring one or more hospitalizations per year. The “Hospital Information System” was used in the follow-up after discharge; patients and their relatives were called every month for six months and information was received. The COPD assessment test (CAT; Appendix A) and Modified Medical Research Board Dyspnea Scale (mMRC; Appendix A) were used to evaluate the patients’ symptoms. The CAT is an eight-item test that evaluates the effects of COPD in a broad framework, including coughing, sputum, respiratory symptoms, sleep quality, fatigue, and confidence in leaving home. The threshold value for the CAT score is determined as 10, while the threshold value for the mMRC is determined as two and patients that are above the threshold value are included in groups B and D according to the GOLD combined COPD classification. Another dyspnea scale that was used, other than the CAT and mMRC, was the Borg dyspnea scale, which is a subjective evaluation that is related to the severity of the patient’s shortness of breath (Appendix A).

The Charlson comorbidity index, which evaluates 19 comorbid diseases, was used to evaluate comorbidities. Points from one to six are given according to the severity of the diseases and the one-year mortality is evaluated (Appendix A).

A combined assessment is the combination of spirometric classification, symptomatic assessment, and exacerbation risk to understand the individual impact of COPD on the patients (Appendix A). According to the combined evaluation system, spirometric classes one to four provide information about the severity of airflow restriction, while groups A to D provide information about the symptom burden and risk of exacerbation that can be used to guide the treatment.

### 2.3. Laboratory Measurements

Blood samples of the patients were taken at the time of admission to the hospital. The hemogram samples were studied with a Sysmex XN-1000 device (Lincolnshire, IL, USA). The NLR was obtained by dividing the neutrophil count by the number of lymphocytes. The serum CRP value was studied using the immune inhibition method on a Beckman Coulter, AU 2700 Plus device, (Carlsbad, CA, USA). The normal CRP range was 0–5 mg/L. The PCT was measured using lateral chromatography and immune turbidimetry.

### 2.4. Statistical Analysis

The sample size was determined by power analysis method. The SPSS 22.0 program was used for statistical analysis. The mean, standard deviation, median lowest–highest, frequency, and ratio values were used in the descriptive statistics of the data. The distribution of the variables was measured using the Kolmogorov–Smirnov test.

The independent sample *t*-test and a Mann–Whitney U test were used for the analysis of the quantitative independent data (hemogram and biochemical blood values, arterial blood gases, pulmonary function test values, EF and PAP values, number of exacerbations in the previous year, length of hospital stay, Borg and mMRC scalas, age, BMI, and the COPD duration).

The chi-squared test was used for the analysis of the qualitative independent data and a Fischer’s test was used when the conditions of the chi-squared test were not met (gender, comorbid diseases, long term oxygen therapy, NIMV use, GOLD stage, Charlson comorbidity index, number of hospital admissions due to exacerbation at follow-up, and the ICU hospitalizations in the previous year).

A Cox regression analysis was used to determine factors affecting the survival. Pearson’s and Spearman’s correlation tests were used for correlation analysis (other values and between combined GOLD classification). A *p*-value of less than 0.05 was considered statistically significant.

Logistic regression analysis was used for procalcitonin, CRP, and NLR (Appendix A)

## 3. Results

The data, including the demographic characteristics of the patients, BMI, smoking status, performance status, distribution according to GOLD stages, and respiratory device use, are presented in Appendix A. When the patients were evaluated according to the GOLD combined assessment classification, 26 (16%) of 160 patients that were included in our study were GOLD Group A, 21 (13%) GOLD Group B, 24 (15%) GOLD Group C, and 89 (56%) GOLD Group D. The values showing statistical significance with the combined GOLD classification were the PAP (r = 0.166 *, *p* < 0.05), PCT (r = 0.165 *, *p* < 0.05) and Charlson comorbidity index (r = 0.163 *, *p* < 0.05). One or more concomitant systemic diseases was present in 110 (68.8%) of the patients that were included in the study. In classifying the accompanying systemic diseases according to the Charlson comorbidity index, the most common ones were hypertension (56%,), diabetes mellitus (31%), congestive heart failure (30%), and ischemic heart disease (30%). Other diseases were benign prostatic hypertrophy (BPH; 13%), arrhythmia (7%), and cerebrovascular accident (CVA; 3%). The hemogram, CRP, NLR, platelet/lymphocyte ratio (PLR), PCT, uric acid, uric acid/creatine ratio, total protein, albumin, arterial blood gas, respiratory function test values, and the echocardiography values are shown in Appendix A. In the six-month follow-up, of the 160 patients that were included in the study, 101 (63.1%) of the patients were admitted to the emergency department due to one or more COPD exacerbations, 55 of them (34.4%) were admitted to the ICU due to one or more COPD exacerbations, and 24 of them (15%) died during the follow-up period. Of the patients who died during follow-up, 12.5% (*n* = 3) were GOLD Group A, 12.5% (*n* = 3) were GOLD Group B, 16.7% (*n* = 4) were GOLD Group C, and 58.3% (*n* = 14) were GOLD Group D. According to their FEV1 values, 12.5% (*n* = 3) were Stage 2, 45.8% (*n* = 11) were Stage 3, and 41.7% (*n* = 10) were Stage 4. When we evaluated their early mortality, the number of deaths in the first month was 3.75% (*n* = 6). Then, 66% (*n* = 4) of those who died in the first month were GOLD Group D, while 33% (*n* = 2) of them were GOLD Group A. When classified according to FEV1, 83% (*n* = 5) of them were Stage 3–4 and 17% of them (*n* = 1) were Stage 2. The patients were divided into groups with or without mortality, with or without an exacerbation, and with or without ICU hospitalization during the six-month follow-up after discharge. In the group with mortality, the age and GOLD stage of the patients were significantly higher than in the non-mortality group (*p* < 0.05). The BMI of the patients in the group with mortality was significantly lower (*p* < 0.05) than in the group without mortality (Table 1). In the mortality group, the hemoglobin (HGB), HCT, and PO2 values were significantly lower (*p* < 0.05) than in the non-mortality group (Table 2). The patients’ age, Borg score, mMRC score, non-invasive mechanical ventilation (NIMV) use, number of exacerbations in the previous year, red cell distribution width (RDW), eosinophil count, and PCO2 values were significantly higher 0 (*p* < 0.05) in the group that was admitted to the ICU at the six-month follow-up (Table 3). In the group with ICU hospitalization, the HCT and EF values were significantly lower (*p* < 0.05) at the six-month follow-up than the group without ICU hospitalization (Table 4). The duration of COPD and the number of exacerbations in the previous year were significantly higher (*p* < 0.05) in the group with at least one hospital admission during follow-up due to COPD exacerbation than in the group without exacerbation (Table 5). The FEV1%, FEV1/FVC, and FVC% values were significantly lower **(*p* < 0.05)** in the group who had at least one hospital admission due to COPD exacerbation during follow-up (Table 6). A Cox regression analysis was used to determine the factors affecting survival (Table 7). In the univariate model, age, BMI, HGB, HCT, NLR, PO2, and gold stage were found to have a significant (*p* < 0.05) efficacy in predicting the survival time. In the multivariate model, age, NLR, PO2, and gold stage were observed to have significant-independent (*p* < 0.05) efficacy in predicting the survival time.

## 4. Discussion

Studies have shown the importance of systemic inflammation in the pathogenesis of COPD, the relationship between the inflammation during the exacerbation and the severity of the disease, and also several inflammatory biomarkers of this inflammation [7,9,10].

Age is considered a risk factor for the development of COPD, and accordingly, the prevalence of COPD increases with age [11]. Reasons such as aggravation of symptoms of COPD patients, a decrease in performance scores, and an increase in mortality risk with increasing age means that elderly patients who present with an exacerbation of COPD should be treated as inpatients rather than outpatients [12]. The reason for the higher average age of our patient population was that only patients that were hospitalized due to COPD exacerbation were included in our study. In addition, parallel to the relationship between age and mortality in the literature, the mean age of our patients was found to be significantly higher in the group with mortality in the six-month follow-up after discharge compared to the group without mortality.

According to the ALPHABET multicentered study that was conducted in Turkey, when categorizing COPD patients according to the GOLD combined COPD classification, Group A had the highest number of patients with 41.1%. Groups B, C, and D then had 20.8%, 13.2%, and 25% of the patients, respectively [13]. On the contrary, in this study, the majority of the patient population were Group D (56%, *n* = 89) patients. This can be explained by the fact that only d patients that were hospitalize due to COPD exacerbation were included in our study, and the risk of hospitalization in advanced stage cases is higher according to the literature [2,3].

According to the GOLD guidelines, the most common comorbid diseases accompanying COPD are cardiovascular diseases (heart failure, ischemic heart disease, arrhythmias, peripheral vascular disease, hypertension), osteoporosis, anxiety and depression, lung cancer, metabolic syndrome and diabetes, gastroesophageal reflux disease, bronchiectasis, and obstructive sleep-apnea syndrome (OSAS) [1]. In parallel with the published studies, we found that the most accompanying diseases were hypertension and cardiovascular diseases in our patient population [14,15,16]. In our study, as the degree of the group increased among the GOLD A, B, C, and D groups, the number of comorbidities and the Charlson comorbidity index, an objective assessment of comorbidity, also increased. This shows that comorbid diseases should be considered as a factor that increases the severity of COPD.

Studies have shown that patients that are hospitalized due to COPD exacerbation have a poor long-term prognosis. Although different numbers have been reported in different studies in the literature, the five-year mortality rate was found to be approximately 50% in the meta-analysis study that was conducted by Hoogendoorn et al., [17]. In a study that was based in the United States, the one-year mortality was 21% and the five-year mortality was 55% [18]. The early mortality rate in the first month of assessment of the patients that were included in our study was 3.75% (*n* = 6), and the mortality rate during the six months of follow-up was 15% (*n* = 24). In our study, 58.3% (*n* = 14) of the patients who died during follow-up were in GOLD Group D and 87.5% (*n* = 21) were at Stage 3 or 4 according to the FEV1. This shows that, in accordance with the literature on this subject [1], mortality is higher in GOLD group D, where the symptom severity, number of exacerbations, and exacerbations requiring hospitalization are higher, and where the mortality increases as the degree of obstruction increases.

In recent years, many inflammatory markers have been investigated to reveal the factors that determine the prognosis in COPD. Since the NLR is an easily accessible, simple, and inexpensive parameter that is obtained by dividing the number of neutrophils by the number of lymphocytes, its possibility to be used in underdeveloped and developing countries has led to many studies and generally yielded worse clinical and mortality-related results. [6,19]. It has been stated that it can be a marker of mortality in cardiac diseases, a strong prognostic factor in various types of cancer, and a marker of inflammatory or infectious pathologies and postoperative complications [19]. In a study that was conducted with intensive care patients, the NLR was found to be useful for determining the severity of the disease and mortality when compared with sepsis scores such as the APACHE II (Acute Physiology and Chronic Health Evaluation II) and SOFA (Sepsis-Related Organ Failure Assessment) [20]. There are studies showing that the NLR value can be used as a prognostic factor due to the pathogenesis of COPD including inflammatory processes and especially increased inflammation during the exacerbation period [6,21,22]. In the study that was conducted by Günay et al., compared to the healthy control group, the mean NLR in patients with stable COPD was significantly higher, and compared to stable COPD patients the mean NLR was significantly higher in COPD patients presenting with exacerbation [23]. In the study that was conducted by Aksoy et al., patients that were presenting with COPD exacerbation were divided into two subgroups—eosinophilic and neutrophilic—and it was found that the mean NLR and CRP values were significantly higher in the neutrophilic group [24]. In the study that was conducted by Saltürk et al., patients that were hospitalized in intensive care due to COPD exacerbation were divided into two groups—eosinophilic and non-eosinophilic—and the NLR, CRP levels, and the length of stay in intensive care were higher in the non-eosinophilic group [25]. Duman et al. also divided the patients who were hospitalized with COPD exacerbation into two groups—non-eosinophilic and eosinophilic—and high NLR and CRP levels were detected in non-eosinophilic patients. It was also shown that the NLR and CRP values decreased in parallel with each other during the follow-up. In the non-eosinophilic group, the length of hospital stay and the number of hospital admissions after discharge were higher [26]. However, data from Csoma et al. do not support an increased risk of earlier recurring moderate or severe relapses in patients that were hospitalized with eosinophilic exacerbations of COPD [27]. In our study, a significant relationship was found between the NLR and the CRP, NLR, and PCT. This relationship, which is in accordance with the findings of other studies, shows that the NLR can guide us in making decisions on the hospitalization and discharge of future patients. In our study, as the NLR increased, the number of admissions to hospital due to COPD exacerbation increased in the six months of follow-up after discharge and NLR was found to be a statistically significant factor in survival in a Cox regression analysis.

Since COPD is a disease that progresses with increased inflammation both in the stable period and during exacerbation periods, the increase in the CRP level was directly associated with COPD and systemic inflammation. Some studies have reported high CRP values in COPD patients even in the stable period [28,29,30], and in some of them, no significant difference was found between the healthy group and the healthy group in the stable period [31]. The reason why serum CRP levels were not increased at the time of the diagnosis of a COPD exacerbation may be the local onset of inflammation [9]. Therefore, the CRP level may be a guide in the follow-up and for the prognosis, rather than diagnosing or excluding the COPD exacerbation. In a study that was conducted by Torres et al., as the CRP level increased in stable COPD patients, a decrease in the arterial oxygen pressure and six-minute walking test distance were found, along with an increase in the degree of obstruction [28]. In a study that was conducted by Diaz et al., a positive correlation was found between the CRP level of COPD patients and their BMI and the number of exacerbations in the last year, and a negative correlation was found with FEV1 and PO2 [32]. In our study, no significant relationship was found between the CRP and the degree of COPD obstruction, FEV1, or BMI, but the relationship between increased hypoxia and CRP was found to be significant, which is consistent with the information that hypoxemia triggers oxidative stress and inflammation in COPD patients [33]. In our study, low PO2, oxygen saturation (Sat O2) and pH values, and high PCO2 values were found in the patient group with high CRP. In the follow-up after discharge, high PCO2 values were found in the group with ICU hospitalization and low PO2 values in the group with mortality. Tofan et al. associated high CRP levels with long hospitalizations in patients that were hospitalized with COPD exacerbation [34]. Similarly, in the study that was conducted by Kawamatawong et al., it was found that increased CRP, and especially increased procalcitonin levels, required longer hospitalization due to COPD exacerbation [35]. In our study, the CRP and procalcitonin levels were found to be correlated with each other and both the CRP and procalcitonin levels were found to be associated with a long hospital stay.

Chronic disease anemia with low circulating HGB levels is an abnormality that occurs in many inflammatory diseases. Although COPD is “traditionally” associated with polycythemia, systemic inflammation, now considered a feature of COPD, is a possible cause of low HGB and HCT levels. If anemia is present in COPD, it may worsen dyspnea and limit exercise tolerance. Anemia has been associated with an increased risk of mortality and exacerbation in COPD patients [36]. In our study, a negative correlation was found between CRP and HCT, which supports the studies indicating that chronic inflammation causes anemia. It was shown that low HGB and HCT values were significantly correlated with mortality, and a correlation was found between low HCT values and the number of ICU hospitalizations at the six-month follow-up.

Studies have shown that the RDW is associated with the disease severity and hospitalization in COPD [37,38,39,40]. In our study, there was a significant correlation between the RDW and ICU admission after discharge.

Inflammatory stimuli lead to a number of systemic changes that are clinically characterized by fever, weakness, and depression, which are called acute-phase reactions. Biochemically, these are characterized by changes in plasma proteins; increasing factors in plasma are called positive acute phase reactants and decreasing ones are called negative acute phase reactants, and albumin is one of the most important negative acute-phase reactants [41]. Albumin is used as a parameter indicating the nutritional status, but Ishida et al. stated in a study they conducted on burn patients, to the contrary, that albumin was not related to the nutritional status but was associated with systemic inflammation and that high CRP and low albumin levels were correlated [42]. Don et al. stated that albumin is associated with both nutrition and inflammation [43]. Using these features, the CRP/albumin ratio has been studied in determining the prognosis of various diseases [4,44,45]. Although there was no significant relationship that was found between albumin and CRP in our study, in the group with high PCT levels, the albumin values were found to be low. This situation supports the studies that have found that the albumin level decreases with inflammation.

COPD is a chronic inflammatory disease with exacerbations, and hypoxemia is triggered especially during these exacerbations. Tissue hypoxemia accelerates anaerobic metabolism and causes the formation of uric acid, the end-product of purine metabolism [10]. Hypoxemia triggers oxidative stress and inflammation in COPD patients [33]. There are studies in which a positive and significant association has been found between serum uric acid and some inflammatory markers such as CRP [46]. However, in a study that was conducted on patients with acute gouty arthritis and bacterial arthritis, the patients’ procalcitonin levels were compared and procalcitonin was found to be higher in bacterial arthritis patients, but not in gouty arthritis where the inflammation and uric acid levels are high [47]. Although there was a positive correlation in the comparison of uric acid, CRP, and procalcitonin in our study, no statistically significant relationship was found. Although it has been reported that there is a relationship between the severity of airflow obstruction, dyspnea, and uric acid level [48], in our study, no relationship was shown between uric acid and the FEV1 value, arterial blood gases, or mMRC score. In our study, two parameters that were significantly associated with increased uric acid levels in COPD patients were the BMI and PAP. In parallel with studies showing that the serum uric acid is high in patients with pulmonary hypertension and that high uric acid is associated with mortality [49,50], in our study, higher PAP values were found in the patient group with uric acid values that were above the median value. This can be explained by the relationship between chronic hypoxia and pulmonary hypertension in COPD. Since the serum uric acid level is dependent on the destruction of purines that are taken with the diet, as well as the destruction of endogenous purines, a relationship has also been established between the diet, BMI, and serum uric acid level [51]. This may explain the positive relationship between the serum uric acid level and BMI in our study.

Although there is a positive correlation between the BMI and mortality, it has been argued that the opposite is valid for COPD, and our study produced results in this direction [52]. Some researchers stated that mortality is associated with a very low muscle ratio from low BMI, and more accurate results can be obtained by evaluating the lean body weight instead of evaluating the total body weight, but since only BMI was looked at in our study, no comparison was made with lean body weight.

Another relationship that was found in our study, apart from the relationship between the PAP and uric acid, was the relationship between the PAP and GOLD classification. It has been reported that the mean pulmonary artery pressure increases with the disease severity in COPD, and high PAP values are associated with hospitalization and low life expectancy [53]. In our study, a positive correlation was found between the GOLD A, B, C, and D classes and the PAP values. This relationship shows the importance of considering pulmonary arterial hypertension as a comorbidity that aggravates COPD.

Studies have reported that there is a weak correlation between the FEV1 and the patient’s symptom level, and so symptomatic evaluation is required in addition to spirometric evaluation [2,3,54]. Although there are studies indicating that the mortality of patients with a high degree of obstruction is higher, this has been proposed only by looking at pulmonary function tests [55,56]. There are also studies indicating that the mortality increases more significantly when evaluated according to the GOLD A, B, C, and D classification [57]. In our study, when the mortality was classified according to the FEV1, it was found to be increased both in the severe and very severe groups and in the GOLD D group. At the same time, it was observed that the FEV1%, FEV1/FVC, and FVC% values were lower in patients who were admitted to the hospital with relapses in the six-month follow-up after discharge compared to patients without exacerbation. This shows that although the symptoms and the history of previous exacerbation and hospitalization are important in the evaluation of COPD patients, pulmonary function tests can be used as an important objective criterion, especially for patients who cannot express themselves adequately or who cannot provide an accurate medical history.

In our study, the number of exacerbations in the previous year was higher in the group that was admitted to intensive care during the follow-up compared to the group that was not. The duration of COPD disease was longer in the group with an exacerbation compared to the group that had not an attack at follow-up. It has been stated that the most important predictor of frequent exacerbations are previous exacerbations [2]. In our study, previous exacerbations were found to be associated with subsequent exacerbations and intensive care admission. The determinants of subsequent exacerbations were the duration of COPD and the number of exacerbations in the previous year, in accordance with the literature.

Heart failure, which is frequently seen as an accompanying comorbid disease in COPD patients, was the most common accompanying chronic disease in our population. Heart failure is associated with mortality as an independent factor in COPD patients and is a factor that can trigger, worsen, and mimic a COPD exacerbation [1]. In our study, although the EF was found to be lower in the group with mortality at the six-month follow-up after discharge, there was no statistically significant difference, but the EF was found to be statistically significantly lower in the group with ICU hospitalization at the six-month follow-up. In the treatment of heart failure accompanying COPD, treatment with NIMV in addition to the traditional treatment has also shown beneficial results [58]. In our study, there was a statistically insignificant relationship between low EF and high mortality. In addition, there was a statistically significant relationship between low EF values and increased ICU need at follow-up.

Limitations: Since only hospitalized patients were included in the study, there was no homogeneous distribution and group D patients constituted the majority according to GOLD staging. In addition, the single-center nature of our study caused the patient population to be selected from a limited environment.

## 5. Conclusions

These findings suggest that advanced age, a long COPD duration, low PFT values, high dyspnea scores, increased inflammatory markers, low BMI, and anemia are the most effective factors in the determining prognosis of patients with COPD attack. It was concluded that scoring combining the selected biomarkers and other factors affecting the prognosis is more powerful in determining the prognosis. We think that more detailed information on this matter can be obtained with new prospective studies that include longer follow-up periods (especially one to five years in terms of mortality) and larger number of patients.

## Figures and Tables

**Table 1 medicina-58-00163-t001:** Comparison of general data between the group with and without mortality at follow-up.

		Mortality (−)				Mortality (+)				*p*	
		Mean ± s.d./*n*–%			Median	Mean ± s.d./*n*–%			Median		
Age		65.4	±	10.2	66.0	72.1	±	11.9	70.5	**0.019 ***	** ^m^ **
Gender	Female	19		14.0%		2		8.3%		0.451	** ^X²^ **
	Male	117		86.0%		22		91.7%			
BMI		26.0	±	4.9	25.0	23.7	±	4.6	23.3	**0.031 ***	** ^m^ **
COPD Duration (years)		7.7	±	7.0	5.5	8.7	±	7.1	8.0	0.373	** ^m^ **
Active Smokers	(+)	117		86.0%		20		83.3%		0.729	** ^X²^ **
	(−)	19		14.0%		4		16.7%			
Comorbid Diseases	(−)	44		32.4%		6		25.0%		0.474	** ^X²^ **
	(+)	92		67.6%		18		75.0%			
BORG Scale		2.5	±	1.2	3.0	2.4	±	1.2	2.5	0.782	** ^m^ **
mMRC Scale		6.4	±	2.6	7.0	6.0	±	2.5	5.5	0.573	** ^m^ **
Long-term oxygen therapy	(−)	16		11.8%		3		12.5%		0.918	** ^X²^ **
	(+)	120		88.2%		21		87.5%			
NIMV	(−)	57		41.9%		9		37.5%		0.686	** ^X²^ **
	(+)	79		58.1%		15		62.5%			
GOLD Stage	I	5		3.7%		0		0.0%		**0.048 ***	** ^X²^ **
	II	43		31.6%		3		12.5%			
	III	63		46.3%		11		45.8%			
	IV	25		18.4%		10		41.7%			
Charlson Comorbidity Index	I	81		59.6%		10		41.7%		0.810	** ^X²^ **
	II	32		23.5%		11		45.8%			
	III	16		11.8%		3		12.5%			
	IV	7		5.1%		0		0.0%			
Number of Hospital Admissions due to Exacerbation at Follow-up	(−)	50		36.8%		9		37.5%		0.945	** ^X²^ **
	(+)	86		63.2%		15		62.5%			
Number of Exacerbations in the Previous Year		3.2	±	3.1	2.0	2.9	±	2.4	2.0	0.665	** ^m^ **
Length of Hospital stay (days)		7.2	±	2.4	7.0	8.3	±	4.5	6.5	0.670	** ^m^ **
ICU Hospitalizations in the Previous Year	(−)	92		67.6%		13		54.2%		0.200	** ^X²^ **
	(+)	44		32.4%		11		45.8%			

Data are presented as the mean ± standard deviation **^m^** Mann–Whitney U test/**^X²^** Chi square test BMI: body mass index, mMRC: Modified Medical Research Council, NIMV: Non-invasive mechanical ventilation, ICU: intensive care unit. ***** Bold text indicates statistical significance with a *p*-value that was less than 0.05.

**Table 2 medicina-58-00163-t002:** Comparison of laboratory data between the group with and without mortality at follow-up.

	Mortality (−)				Mortality (+)				*p*	
	Mean ± s.d./*n*–%			Median	Mean ± s.d./*n*–%			Median		
HGB(g/dL)	14.1	±	2.2	14.1	13.0	±	1.9	13.1	**0.014 ***	** ^m^ **
HCT (%)	43.9	±	6.5	43.6	41.0	±	5.7	40.8	**0.031 ***	** ^m^ **
WBC (10^3^/µL)	14.7	±	19.1	11.1	22.5	±	49.3	11.6	0.909	** ^m^ **
Neutrophil (10^3^/µL)	9054	±	4481	7810	10353	±	7197	8495	0.839	** ^m^ **
PLT (10^3^/µL)	25.5	±	10.8	22.9	26.4	±	7.3	25.8	0.336	** ^m^ **
Pct (%)	0.2	±	0.1	0.2	0.2	±	0.1	0.3	0.473	** ^m^ **
PDW	11.3	±	2.0	10.9	10.7	±	1.6	10.0	0.125	** ^m^ **
MPV (fL)	9.8	±	1.0	9.7	9.5	±	0.9	9.2	0.183	** ^m^ **
RDW-CV (%)	15.9	±	2.9	14.9	16.3	±	2.3	15.9	0.168	** ^m^ **
Eosinophil (10^3^/µL)	138.5	±	200.5	80.0	191.3	±	313.3	100.0	0.442	** ^m^ **
Eosinophil (%)	1.3	±	1.8	0.7	1.7	±	2.0	0.9	0.308	** ^m^ **
Lymphocyte (10^3^/µL)	1614	±	914	1470	1518	±	831	1410	0.633	** ^m^ **
NLR	7.2	±	5.2	5.9	12.0	±	21.5	5.3	0.861	** ^m^ **
PLR	204	±	167	166	234	±	193	181	0.465	** ^m^ **
CRP (mg/L)	70.8	±	70.5	46.4	72.0	±	82.6	52.7	0.989	** ^m^ **
Procalcitonin (ng/mL)	0.7	±	0.7	0.5	1.1	±	2.2	0.4	0.811	** ^m^ **
Uric Acid (mg/dL)	5.6	±	1.8	5.6	6.1	±	2.0	6.1	0.314	** ^m^ **
Uric Acid/Creatinin	7.0	±	2.2	6.9	6.8	±	1.8	6.5	0.648	** ^m^ **
Total Protein (g/dL)	7.3	±	6.5	6.7	6.8	±	0.7	6.9	0.850	** ^m^ **
Albumin (g/dL)	3.9	±	3.1	3.6	3.6	±	0.6	3.5	0.345	** ^m^ **
pH	7.4	±	0.1	7.4	7.4	±	0.1	7.4	0.101	** ^m^ **
PCO2 (mmHg)	51.7	±	14.4	51.6	53.9	±	17.8	58.5	0.544	** ^m^ **
PO2 (mmHg)	86.3	±	70.9	62.4	63.1	±	30.0	54.2	**0.032 ***	** ^m^ **
O2 Saturation (%)	89.6	±	9.1	90.0	85.5	±	11.4	89.0	0.062	** ^m^ **
FEV1 (L)	1.2	±	2.1	1.0	1.0	±	0.4	0.9	0.690	** ^m^ **
FEV1 (%)	69.7	±	370.9	34.3	37.2	±	15.8	36.0	0.722	** ^m^ **
FVC (L)	2.3	±	5.0	1.6	1.8	±	0.7	1.7	0.903	** ^m^ **
FVC (%)	47.8	±	16.7	46.1	48.2	±	19.1	44.1	0.835	** ^m^ **
FEV1/FVC	59.0	±	13.1	56.6	57.7	±	13.7	55.8	0.760	** ^m^ **
EF (%)	55.8	±	7.3	60.0	53.2	±	10.3	57.5	0.249	** ^m^ **
PAP (mmHg)	35.0	±	13.1	30.0	37.7	±	12.9	32.5	0.371	** ^m^ **

Data are presented as the mean ± standard deviation. **^m^** Mann–Whitney U test HGB: Hemoglobin, HCT: hematocrit, WBC: white blood cells, PLT: platelet, Pct: plateletcrit, PDW: platelet distribution width, MPV: mean platelet volume, RDW-CV: RDW-CV: erythrocyte distribution width—coefficient of variation, NLR: neutrophil lymphocyte ratio, PLR:Platelet lymphocyte ratio, FEV1: forced expiratory volume in first second, FVC: forced vital capacity, EF: ejection fraction, PAP: pulmonary arterial pressure. ***** Bold text indicates statistical significance with a *p*-value less than 0.05.

**Table 3 medicina-58-00163-t003:** Comparison of the general data of the group with and without ICU admission during follow-up.

		ICU Admission (−)				ICU Admission (+)				*p*	
		Mean ± s.d./*n*–%			Median	Mean ± s.d./*n*–%			Median		
Age		68.2	±	10.2	68.0	63.0	±	10.8	62.0	**0.002 ***	** ^m^ **
Gender	Female	13		8.1%		8		14.5%		0.700	** ^X²^ **
	Male	92		57.5%		47		85.5%			
BMI		25.7	±	5.2	24.9	25.6	±	4.5	24.2	0.801	** ^m^ **
COPD duration (years)		7.8	±	7.2	5.0	8.0	±	6.8	7.0	0.722	** ^m^ **
Active Smokers	(+)	87		54.4%		50		90.9%		0.168	** ^X²^ **
	(−)	18		11.3%		5		9.1%			
Comorbid Diseases	(−)	35		21.9%		15		27.3%		0.432	** ^X²^ **
	(+)	70		43.8%		40		72.7%			
BORG Scale		2.3	±	1.2	2.0	2.7	±	1.2	3.0	**0.018 ***	** ^m^ **
mMRC Scale		5.9	±	2.6	6.0	7.0	±	2.5	8.0	**0.007 ***	** ^m^ **
Long-Term Oxygen Therapy	(−)	15		9.4%		4		7.3%		0.193	** ^X²^ **
	(+)	90		56.3%		51		92.7%			
NIMV	(−)	53		33.1%		13		23.6%		**0.001 ***	** ^X²^ **
	(+)	52		32.5%		42		76.4%			
GOLD Stage	I	4		2.5%		1		1.8%		0.408	** ^X²^ **
	II	32		20.0%		14		25.5%			
	III	50		31.3%		24		43.6%			
	IV	19		11.9%		16		29.1%			
Charlson Comorbidity Index	I	63		60.0%		28		50.9%		0.728	** ^X²^ **
	II	26		24.8%		17		30.9%			
	III	12		11.4%		7		12.7%			
	IV	4		3.8%		3		5.5%			
Number of Hospital Admissions due to Exacerbation at Follow-up	(−)	44		27.5%		15		27.3%		0.068	** ^X²^ **
	(+)	61		38.1%		40		72.7%			
Number of Exacerbations in the Previous Year		2.6	±	2.5	2.0	4.1	±	3.7	3.0	**0.010 ***	** ^m^ **
Length of Hospital Stay (days)		7.3	±	2.3	7.0	7.5	±	3.6	7.0	0.494	** ^m^ **

Data are presented as the mean ± standard deviation **^m^** Mann–Whitney U test/**^X²^** Chi square test BMI: body mass index, mMRC: Modified Medical Research Council, NIMV: non-invasive mechanical ventilation, ICU: intensive care unit ***** Bold text indicates a statistical significance with a *p*-value that was less than 0.05.

**Table 4 medicina-58-00163-t004:** Comparison of laboratory data of the group with and without ICU admission during follow-up.

	ICU Admission (−)				ICU Admission (+)				*p*	
	Mean ±s.d./*n*–%			Median	Mean ±s.d./*n*–%			Median		
HGB (g/dL)	13.8	±	2.2	14.0	14.2	±	2.1	14.0	0.461	** ^m^ **
HCT (%)	45.2	±	6.4	44.2	42.6	±	6.2	42.4	**0.025 ***	** ^m^ **
WBC (10^3^/µL)	16.8	±	30.8	10.8	14.1	±	11.7	11.7	0.233	** ^m^ **
Neutrophil (10^3^/µL)	9002	±	5342	7640	9719	±	4194	8880	0.130	** ^m^ **
PLT (10^3^/µL)	24.7	±	9.3	22.8	27.6	±	12.0	25.5	0.095	** ^m^ **
Pct (%)	0.2	±	0.1	0.2	0.3	±	0.1	0.2	0.131	** ^m^ **
PDW	11.0	±	1.9	10.8	11.5	±	2.0	10.8	0.161	** ^m^ **
MPV (fL)	9.7	±	1.0	9.6	9.9	±	0.9	9.7	0.256	** ^m^ **
RDW-CV (%)	15.4	±	2.3	14.8	16.9	±	3.3	15.8	**0.003 ***	** ^m^ **
Eosinophil (10^3^/µL)	141.8	±	247.9	70.0	155.1	±	157.8	110.0	**0.049 ***	** ^m^ **
Eosinophil (%)	1.3	±	2.0	0.6	1.5	±	1.5	1.0	0.150	** ^m^ **
Lymphocyte (10^3^/µL)	1566	±	943	1400	1663	±	818	1480	0.389	** ^m^ **
NLR	8.4	±	11.4	5.5	7.0	±	4.7	5.9	0.947	** ^m^ **
PLR	202	±	138	164	222	±	221	176	0.904	** ^m^ **
CRP (mg/L)	73.6	±	74.5	49.0	65.9	±	68.0	42.8	0.689	** ^m^ **
Prokalsitonin (ng/mL)	0.7	±	1.2	0.5	0.7	±	0.8	0.4	0.737	** ^m^ **
Uric Acid (mg/dL)	5.8	±	1.8	5.7	5.5	±	2.0	5.3	0.222	** ^m^ **
Uric Acid/Creatinine	7.0	±	2.2	6.8	7.0	±	2.2	6.5	0.779	** ^m^ **
Total Protein (g/dL)	7.5	±	7.4	6.7	6.8	±	0.7	6.8	0.682	** ^m^ **
Albumin (g/dL)	3.6	±	0.4	3.6	4.2	±	4.8	3.5	0.489	** ^m^ **
pH	7.4	±	0.1	7.4	7.4	±	0.1	7.4	0.072	** ^m^ **
PCO2 (mmHg)	49.6	±	12.8	48.6	56.8	±	17.3	58.8	**0.007 ***	** ^m^ **
PO2 (mmHg)	61.9	±	25.2	55.0	75.6	±	57.0	56.3	0.781	** ^m^ **
O2 Saturation (%)	87.3	±	8.3	89.1	84.0	±	15.1	89.5	0.990	** ^m^ **
FEV1 (L)	1.3	±	2.4	0.9	1.0	±	0.4	1.0	0.596	** ^m^ **
FEV1 (%)	38.6	±	14.4	35.2	114.9	±	583.2	33.9	0.369	** ^m^ **
FVC (L)	2.5	±	5.7	1.7	1.7	±	0.7	1.6	0.341	** ^m^ **
FVC (%)	49.1	±	16.4	47.9	45.6	±	18.2	42.0	0.139	** ^m^ **
FEV1/FVC	59.6	±	13.8	58.2	57.3	±	11.8	55.8	0.460	** ^m^ **
EF (%)	56.5	±	6.6	60.0	53.3	±	9.5	57.5	**0.032**	** ^m^ **
PAP (mmHg)	34.6	±	11.8	30.0	37.1	±	15.2	30.0	0.710	** ^m^ **

Data are presented as the mean ± standard deviation. **^m^** Mann–Whitney U test HGB: hemoglobin, HCT: hematocrit, WBC: white blood cells, PLT: platelet, Pct: plateletcrit, PDW: platelet distribution width, MPV: mean platelet volume, RDW-CV: RDW-CV: erythrocyte distribution width—coefficient of variation, NLR: neutrophil lymphocyte ratio, PLR: platelet lymphocyte ratio, FEV1: forced expiratory volume in first second, FVC: forced vital capacity, EF: ejection fraction, PAP: pulmonary arterial pressure, ICU: intensive care unit ***** Bold text indicates statistical significance with a *p*-value that was less than 0.05.

**Table 5 medicina-58-00163-t005:** Comparison of the general data of the group with at least one hospital admission due to COPD exacerbation at the follow-up and the group with no exacerbation.

		Exacerbation (−)				Exacerbation (+)				*p*	
		Mean ± s.d./*n*–%			Median	Mean ± s.d./*n*–%			Median		
Age		68.2	±	10.4	68.0	65.4	±	10.8	65.0	0.158	** ^m^ **
Gender	Female	5		8.5%		16		15.8%		0.183	** ^X²^ **
	Male	54		91.5%		85		84.2%			
BMI		25.7	±	4.8	24.2	25.7	±	5.0	24.9	0.886	** ^m^ **
COPD Duration (years)		6.0	±	5.8	5.0	9.0	±	7.4	7.0	**0.011 ***	** ^m^ **
Active smokers	(+)	50		84.7%		87		86.1%		0.809	** ^X²^ **
	(−)	9		15.3%		14		13.9%			
Comorbid Diseases	(−)	18		30.5%		32		31.7%		0.877	** ^X²^ **
	(+)	41		69.5%		69		68.3%			
BORG Scale		2.3	±	1.2	2.0	2.6	±	1.2	3.0	0.093	** ^m^ **
mMRC Scale		6.0	±	2.9	6.0	6.5	±	2.4	7.0	0.428	** ^m^ **
Long-Term Oxygen Therapy	(−)	7		11.9%		12		11.9%		0.997	** ^X²^ **
	(+)	52		88.1%		89		88.1%			
NIMV	(−)	24		40.7%		42		41.6%		0.911	** ^X²^ **
	(+)	35		59.3%		59		58.4%			
GOLD Stage	I	2		3.4%		3		3.0%		0.588	** ^X²^ **
	II	20		33.9%		26		25.7%			
	III	27		45.8%		47		46.5%			
	IV	10		16.9%		25		24.8%			
Charlson Comorbidity Index	I	30		50.8%		61		60.4%		0.299	** ^X²^ **
	II	21		35.6%		22		21.8%			
	III	6		10.2%		13		12.9%			
	IV	2		3.4%		5		5.0%			
Length of Hospital Stay (days)		7.5	±	2.7	7.0	7.3	±	2.9	7.0	0.600	** ^m^ **
Number of Exacerbations in the Previous Year		2.8	±	2.7	2	4.0	±	3.7	3	**0.018 ***	** ^m^ **

Data are presented as the mean ± standard deviation **^m^** Mann–Whitney U test/**^X²^** Chi square test BMI: body mass index, mMRC: Modified Medical Research Council, NIMV: non-invasive mechanical ventilation, ICU: intensive care unit. ***** Bold text indicates statistical significance with a *p*-value that was less than 0.05.

**Table 6 medicina-58-00163-t006:** Comparison of the laboratory data of the group with at least one hospital admission due to COPD exacerbation during follow-up and the group with no exacerbation.

	Exacerbation (−)				Exacerbation (+)				*p*	
	Mean ± s.d./*n*–%			Median	Mean ± s.d./*n*–%			Median		
HGB (g/dL)	13.8	±	2.2	13.3	14.0	±	2.1	14.1	0.329	** ^m^ **
HCT (%)	42.8	±	7.0	41.8	43.9	±	6.0	44.0	0.100	** ^m^ **
WBC (10^3^/µL)	15.2	±	31.6	10.8	16.3	±	22.0	11.7	0.177	** ^m^ **
Neutrophil (10^3^/µL)	8086	±	3720	7180	9928	±	5484	8440	0.094	** ^m^ **
PLT (10^3^/µL)	24.7	±	8.7	23.2	26.3	±	11.2	22.9	0.776	** ^m^ **
Pct (%)	0.2	±	0.1	0.2	0.2	±	0.1	0.2	0.945	** ^m^ **
PDW	11.2	±	2.0	10.7	11.1	±	1.9	10.9	0.758	** ^m^ **
MPV (fL)	9.7	±	1.0	9.6	9.7	±	1.0	9.7	0.885	** ^m^ **
RDW-CV (%)	15.6	±	2.5	15.0	16.1	±	2.9	15.1	0.452	** ^m^ **
Eosinophil (10^3^/µL)	156.7	±	297.5	50.0	140.4	±	161.2	90.0	0.101	** ^m^ **
Eosinophil (%)	1.4	±	2.0	0.6	1.4	±	1.8	0.9	0.145	** ^m^ **
Lymphocyte (10^3^/µL)	1604	±	1136	1350	1597	±	736	1490	0.524	** ^m^ **
NLR	7.2	±	7.1	5.4	8.3	±	10.8	5.9	0.678	** ^m^ **
PLR	222	±	214	165	201	±	140	171	0.749	** ^m^ **
CRP (mg/L)	69.1	±	73.4	33.2	72.0	±	71.8	53.4	0.918	** ^m^ **
Procalcitonin (ng/mL)	0.8	±	1.5	0.5	0.7	±	0.7	0.5	0.787	** ^m^ **
Uric Acid (mg/dL)	5.9	±	1.9	5.7	5.5	±	1.8	5.6	0.277	** ^m^ **
Uric Acid/Creatinine	6.9	±	2.3	6.6	7.0	±	2.1	6.9	0.557	** ^m^ **
Total Protein (g/dL)	8.0	±	9.9	6.6	6.8	±	0.6	6.8	0.777	** ^m^ **
Albumin (g/dL)	3.6	±	0.5	3.5	4.0	±	3.6	3.6	0.423	** ^m^ **
pH	7.4	±	0.1	7.4	7.4	±	0.1	7.4	0.977	** ^m^ **
PCO2 (mmHg)	51.0	±	14.7	51.2	52.6	±	15.0	52.0	0.661	** ^m^ **
PO2 (mmHg)	62.1	±	28.1	54.7	69.2	±	44.8	55.3	0.760	** ^m^ **
O2 Saturation (%)	87.5	±	7.3	89.1	85.4	±	12.9	89.6	0.843	** ^m^ **
FEV1 (L)	1.6	±	3.2	1.1	1.0	±	0.4	0.9	**0.009 ***	** ^m^ **
FEV1 (%)	114.9	±	562.4	40.5	35.6	±	13.5	32.7	**0.021 ***	** ^m^ **
FVC (L)	3.2	±	7.6	1.9	1.6	±	0.7	1.5	**0.017 ***	** ^m^ **
FVC (%)	52.2	±	18.6	50.3	45.4	±	15.6	43.0	**0.024 ***	** ^m^ **
FEV1/FVC	60.5	±	13.0	60.5	57.9	±	13.2	55.8	0.240	** ^m^ **
EF (%)	55.2	±	7.4	60.0	55.6	±	8.1	60.0	0.296	** ^m^ **
PAP (mmHg)	34.6	±	11.7	30.0	35.9	±	13.9	30.0	0.862	** ^m^ **

Data are presented as the mean ± standard deviation. **^m^** Mann–Whitney U test HGB: hemoglobin, HCT: hematocrit, WBC: white blood cells, PLT: platelet, Pct: plateletcrit, PDW: platelet distribution width, MPV: mean platelet volume, RDW-CV: RDW-CV: erythrocyte distribution width—coefficient of variation, NLR: neutrophil lymphocyte ratio, PLR: platelet lymphocyte ratio, FEV1: forced expiratory volume in first second, FVC: forced vital capacity, EF: ejection fraction, PAP: pulmonary arterial pressure. ***** Bold text indicates statistical significance with a *p*-value that was less than 0.05.

**Table 7 medicina-58-00163-t007:** Cox Regression Analysis.

	Univariate Model					Multivariate Model				
	HR	% 95 CI			*p*	HR	% 95 CI			*p*
Age	1.062	1.02	-	1.106	**0.003 ***	1.056	1.015	-	1.099	**0.008**
Gender	1.703	0.4	-	7.243	0.471					
BMI	0.908	0.83	-	0.994	**0.037 ***					
Active smokers	1.23	0.42	-	3.598	0.706					
COPD duration	1.017	0.965	-	1.072	0.533					
Comorbid diseases	1.371	0.544	-	3.453	0.504					
Number of exacerbations in the previous year	0.974	0.844	-	1.124	0.714					
Long term oxygen therapy	0.917	0.274	-	3.076	0.889					
NIMV	1.145	0.501	-	2.617	0.748					
Hemoglobin	0.816	0.688	-	0.968	**0.019 ***					
Hematocrite	0.939	0.884	-	0.997	**0.039 ***					
WBC	1.006	0.996	-	1.015	0.223					
Neutrophile	1	1	-	1	0.219					
Leukocyte	1	1	-	1	0.223					
PLT	1.781	0.704	-	4.504	0.223					
Pct	1.601	0.02	-	130.92	0.834					
PDW	0.844	0.666	-	1.068	0.158					
MPV	0.774	0.505	-	1.185	0.238					
RDW-CV	1.041	0.913	-	1.186	0.549					
Eosinophile count	1.001	0.999	-	1.002	0.256					
Eosinophile (%)	1.092	0.928	-	1.284	0.29					
Lymphocyte	1	0.999	-	1	0.609					
NLR	1.034	1.01	-	1.058	**0.004 ***	1.025	1	-	1.05	**0.049**
PLR	1.001	0.999	-	1.003	0.403					
CRP	1	0.995	-	1.006	0.975					
Procalcitonin	1.187	0.977	-	1.441	0.084					
Uric acid	1.146	0.928	-	1.415	0.205					
Uric acid/Creatine	0.959	0.798	-	1.152	0.653					
Total Protein	0.968	0.783	-	1.195	0.759					
Albumin	0.845	0.433	-	1.651	0.622					
pH	0.004	0	-	1.493	0.068					
PCO2	1.007	0.981	-	1.034	0.583					
PO2	1.007	1.002	-	1.013	**0.012 ***	1.006	1	-	1.012	**0.042**
O2 Sat.	1.039	0.991	-	1.088	0.113					
FEV1	0.813	0.328	-	2.013	0.654					
FEV1 %	0.998	0.974	-	1.023	0.886					
FVC	0.957	0.785	-	1.167	0.663					
FVC %	1.002	0.978	-	1.026	0.884					
FEV1/FVC	0.994	0.964	-	1.026	0.726					
GOLD Stage	2.219	1.256	-	3.92	**0.006 ***	1.879	1.042	-	3.388	**0.036**
***mMRC Scale**	0.944	0.673	-	1.323	0.737					
Borg Scale	0.958	0.823	-	1.115	0.58					
Charlson Comorbidity Index	1.095	0.705	-	1.702	0.686					
EF	0.969	0.93	-	1.008	0.121					
PAP	1.013	0.985	-	1.041	0.363					
Hospitalization in the ICU	1.655	0.741	-	3.694	0.219					
Length of stay in the ICU	1.158	0.905	-	1.482	0.244					
Length of hospital stay	1.11	0.996	-	1.237	0.059					

* Bold text indicates statistical significance with a *p*-value that was less than 0.05.

## Data Availability

The dataset that supports the findings of this study is available on request from the corresponding author.

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
