# Peer review of "What Are the Most Effective Factors in Determining Future Exacerbations, Morbidity Weight, and Mortality in Patients with COPD Attack?"

_medicina, 2022, doi:10.3390/medicina58020163_

Round 1

Reviewer 1 Report

I have read the article by Koc and Sahin with great interest. The authors evaluated a number of variables and correlated them with mortality, ICU admission and future exacerbation risk in patients hospitalised with COPD exacerbation.

Comments:

  • Line 37. You may consider adding https://pubmed.ncbi.nlm.nih.gov/32547001/ which is one of the largest study evaluating FEV1 as a risk factor for exacerbation and mortality. This article may also be important when discussing the results.
  • Statistical analysis. As it was a prospective study, please provide a priori sample size calculations.
  • Please, divide them into different paragraphs and label them with subheadings according to the topics.
  • A few times you mention significant results, yet the accompanying p value is >0.05. Please, clarify.
  • Table 3A. I believe Takipte Atak and KOAH Süresi are in Turkish. Please, use English.
  • Please check the correct concentrations in the tables (i.e. WBC).
  • Line 241. You may consider citing https://pubmed.ncbi.nlm.nih.gov/30008015/ a large study evaluating comorbidities along GOLD grades.
  • Discussion. You may consider citing a similar study to yours https://pubmed.ncbi.nlm.nih.gov/33585654/ which investigated different factors (focus on eosinophils) to predict future exacerbations in patients hospitalised with COPD exacerbation.

Author Response

REVISION LETTER FOR REVIEWER 1

  • The article you suggested has been added to line 37 as a 4th reference.

(Bikov A, Lange P, Anderson JA, Brook RD, Calverley PMA, Celli BR, Cowans NJ, Crim C, Dixon IJ, Martinez FJ, Newby DE, Yates JC, Vestbo J. FEV1 is a stronger mortality predictor than FVC in patients with moderate COPD and with an increased risk for cardiovascular disease. Int J Chron Obstruct Pulmon Dis. 2020 May 20;15:1135-1142. doi: 10.2147/COPD.S242809)

·         “Sample size was determined by power analysis method.” This sentence has been added to the statistical analysis section.·         The statistical analyzes were specified according to the topics and divided into paragraphs.·          “In the group with mortality, the age and GOLD stage of the patients were significantly higher than the non-mortality group (p ˃ 0.05). The BMI of the patients in the group with mortality was significantly lower (p ˃ 0.05) than the group without mortality [Table 1A].” The p>0.05 value in the sentences was fixed as p<0.05.

  •  “Takipte Atak ve KOAH Süresi”in Table 3A has been corrected in English.

·         WBC, neutrophil, platelet, eosinophil and lymphocyte units in the tables were corrected (103/µL).·         The article you suggested has been added to line 241. As 16.reference. (Bikov A, Horváth A, Tomisa G, Bártfai L, Bártfai Z. Changes in the Burden of Comorbidities in Patients with COPD and Asthma-COPD Overlap According to the GOLD 2017 Recommendations. Lung. 2018 Oct;196(5):591-599. doi: 10.1007/s00408-018-0141-7).·         The article you suggested has been added to the discussion section as 27.reference.

(Csoma B, Bikov A, Tóth F, Losonczy G, Müller V, Lázár Z. Blood eosinophils on hospital admission for COPD exacerbation do not predict the recurrence of moderate and severe relapses. ERJ Open Res. 2021 Feb 8;7(1):00543-2020. doi: 10.1183/23120541.00543-2020).

      Revisions made within the article are indicated in bold red and underlined.

Reviewer 2 Report

Thank you for giving me the opportunity to review this article. In this article, the authors aimed to investigate the effectiveness of inflammatory biomarkers and other factors in predicting various clinical conditions in patients hospitalized for COPD exacerbation. The study was planned prospectively and included 160 patients.

My comments and suggestions are below.

  1. I suggest changing the title and shortening it.
  2. The article should be revised with regard to English grammar and spelling errors. (for example, some Turkish words in table 3A should be corrected). Particular attention should be paid to the places where commas should be used. In addition, many English corrections are required. For example, "patients with malignancy" should be used instead of "Cancer patients".
  3. Words to be corrected: Prospetıve, sevrity, aist
  4. The sentence on lines 83-85 should be changed and referenced.
  5. line 137; please replace "significant correlation" with "statistically significant". If correlation is applied, please enter r values together.
  6. Purpose of the study: to investigate the effectiveness of inflammatory biomarkers. However, when you look at it, it is seen that many non-purpose parameters were evaluated in your study. Therefore, I suggest simplifying the article for your purpose. For example, EF and PAP are not inflammatory biomarkers.
  7. I also recommend applying logistic regression for your important parameters in the study.
  8. Please delete any off-purpose parameters in the tables and focus on systemic inflammation parameters so as not to confuse readers.
  9. In one paragraph of discussion, explain the factors that limit your work. Also, please mention the contribution of your findings to our daily practice.
  10. The discussion part of the article is overall well done, thank you for that. However, as I mentioned in the first 9 points, I recommend you to fully focus on your purpose.
  11. More than 50% of references are from the last 5 years, are current and acceptable.

Author Response

REVISION LETTER FOR REVIEWER 2

1.Based on your suggestions, the title is “What are the Most Effective Factors in Determining Future Exacerbations, Morbidity Weight and Mortality in Patients with COPD Attack?” abbreviated as.

2.The article was reviewed in terms of grammar and spelling errors in English. Turkish words in Table 3A have been corrected in English.

3.The spelling of the words you specified has been corrected.

4. This sentence is taken from the GOLD 2021 manual and has a reference at the end of the sentence (reference 1).

5.Changed "significant correlation" to "statistically significant" on line 137.

6.In the aim of the study, which we mentioned in the sentence between lines 53-56 in the introduction, the effect of inflammatory biomarkers and other factors (duration of hospitalization, future attacks, intensive care admissions and mortality) on the prognosis of patients hospitalized with COPD exacerbations  was expressed.

7.Logistic regression analysis was performed for CRP, procalcitonin and NLR. Tables added as supplemental file (Supplementary Table 6-7-8).

8.Other factors such as arterial blood gas, PFT (Pulmonary function test), EF (Ejection fraction), PAP (Pulmonaryarterial pressure) values, body mass index (BMI), Charlson Comorbidity Index, mMRC (Modified Medical Research Council) and Borg scale are also in parentheses added to the sentence in the between lines 53-56 in the introduction.

9. Sentences indicating limitations have been added to the discussion section in article.“Limitations: Because only hospitalized patients were included in the study, there was no homogeneous distribution and group D patients constituted the majority according to GOLD staging. In addition, the single-center nature of our study caused the patient population to be selected from a limited environment. As we stated in the conclusion part of our article, the combination of inflammatory markers, advanced age, advanced dyspnea parameters and low PFT findings, long disease duration, low BMI and anemia are indicators of poor prognosis in outpatients with COPD. This will be beneficial to our daily practice in terms of the course of the disease.

10. Thank you for your comment. However, since the aim of our article is to examine not only inflammatory biomarkers, but also other factors, as mentioned above, a detailed study was conducted.

11. Thank you for your comment. Also added in the article are new references 4, 16, and 27.

Our article has been revised in terms of English grammar.

 Revisions made within the article are indicated in bold red and underlined.

Reviewer 3 Report

The author try to find some inflammatory associated biomarkers and factors to predict the future risk of AE and mortality by using parameters obtained from admitted patients with AECOPD.

  1. However, the most important factors (such as NLR, CRP, PLR, WBC, Neutrophil, PCT etc.) showed negative findings. So the primary aim did not accomplish.
  2. Since this cohort has been followed up for at least 6 months, cox-regression model could be used to evaluate the time effect. The statisitc part in the presented tables only using Mann-whitney u test and X² Chi square test, therefore we cannot understand the time effect of these parameters.
  3. Other factors showed positive associated with recurrent AE in the current study (e.g Poor FEV1, frequently AE, old age, mMRC, BORG scale) were well known. Therefore, the study did not provide more new information for the readers or physicians to improve their care flow in patients with COPD.
  4. Extensive editing of English language and style required.

Author Response

REVISION LETTER FOR REVIEWER 3

1.Our article in 53-56. between the lines “In our study we aimed to investigate the efficacy of inflammatory biomarkers and other factors to predict the frequency of acute attacks, duration of hospitalization, morbidity and mortality in hospitalized patients due to COPD exacerbation.” As it can be understood from the sentence, we evaluate both inflammatory biomarkers and other factors such as arterial blood gas, PFT (Pulmonary function test), EF (Ejection fraction), PAP (Pulmonary arterial pressure) values, body mass index (BMI), Charlson Comorbidity Index, MMRC. (Modified Medical Research Council) and BORG scales. Our main goal was to examine all the factors that we mentioned. NLR, CRP, PLR, WBC, Neutrophil, PCT etc. negative prognosis is also an important result for us. Other factors (low BMI, anemia, low PFT, high dyspnea scores, advanced age and long duration of COPD) have been found to have significant effects on prognosis. Taken together, these are important results.

  1. Cox regression analysis was made and added to the article as Table 4.
  2. Our article is valuable as a prospective study. In addition to these, low BMI and anemia were also associated with mortality.
  3. Our article has been revised in terms of English grammar.
  4. Revisions made within the article are indicated in bold red and underlined.

Reviewer 4 Report

In this study the Authors aimed to detect the factors that influence the prognosis in hospitalized COPD patients. These encompass very well known factors, such as the age of the patients, results of pulmonary function tests and arterial blood gas studies, as well as the results  of some laboratory data, including biochemical indices of inflammation. Thus there is no novel findings. However, the value of the study is associated with the fact that all these commonly known factors have been studied in one relatively homogenous group of patients. Although this is prospective study  (and that is its value) the time of observation (6 months) is relatively short.

The manuscript needs some corrections. All over the text some editorial corrections are needed, including English grammar and more clear construction of the sentences. Regarding the proper use of abbreviations:  in two different places of the manuscript the abbreviation mMRC has been explained in two different ways (one of them erroneously). MATERIAL: needs adding some more information about comorbidities: surprisingly there were no patients with sleep apnoea – according to the global statistics there should be  a significant number of patients with this comorbidity: RESULTS: In the Table 1B among “laboratory  data” pulmonary function tests and the results of  arterial blood gas studies can be found. DISCUSSION: the  paragraph  related to BMI as prognostic factor  needs enlargement (possibly with the comment on obesity paradox – if any in this study). CONCLUSION: this part needs profound changes. Its first sentence: when read separately carries no information; second sentence: is not related to the present study, as the Authors mention new biomarkers what has not been the aim of this study; the third sentence seems to be repetition of incomplete information from the first sentence

REFERENCES: some more recent studies and some editorial corrections are needed.   Among 55 citations there are three bibliographical positions published in 2019, all the rest  have been published much earlier,  with the exception of the recent citation regarding the definition of COPD; some of the titles of the journals are in full version and some of them – in abbreviated – should  be uniform and accordingly to the editorial rules DATA AVAILABILITY STATEMENT has not been  reported (with erroneous explanation)

Author Response

REVISION LETTER FOR REVIEWER 4

In our study, we decided that our observation period would be 1 year. We were getting information about the patients by phone every month. However, since some of the patients could not be reached after 6 months, we decided to have a follow-up period of 6 months.

Our article has been revised in English. Mistake of in mMRC has been corrected.

MATERIAL: The number of our patients with sleep-apnea was two. However, it was not specified because it was statistically low (2/160=0.012).

RESULTS: The results of pulmonary function tests (FEV1, FVC, FEV1/FVC) and arterial blood gases (pH, PO2, PCO2, O2 saturation) are already shown among the “laboratory data” in Table 1B.

DISCUSSION: “Some researchers stated that mortality is associated with a very low muscle ratio from low BMI, and more accurate results can be obtained by evaluating lean body weight instead of evaluating total body weight, but since only BMI was looked at in our study, no comparison was made with lean body weight. ” This has been added to the discussion as additional information in article.

CONCLUSION: First sentence “These findings suggest that advanced age, long COPD duration, low PFT values, high dyspnea scores, increased inflammatory markers, low BMI and anemia are the most effective factors in the determining prognosis of patients with COPD attack.” changed to.

 The second sentence was removed from the article.

The third sentence is left the same.

“We think that more detailed information on this matter can be obtained with new prospective studies that include longer follow-up periods (especially 1 year, 5 years in terms of mortality) and larger number of patients.” sentence added.

REFERENCES: Three new article (4., 16. and 27.references) have been added. References have been corrected according to the spelling rules of the journal.

DATA AVAILABILITY STATEMENT: Corrected as “The dataset that supports the findings of this study is available on request from the corresponding author”.

Revisions made within the article are indicated in bold red and underlined.

Round 2

Reviewer 1 Report

I am happy with the changes and suggest acceptance.

Author Response

Thank you very much

Reviewer 3 Report

  1. Suggest change the title to " What are the Most Effective Factors in Determining Future risk of exacerbation and Mortality in Patients with COPD Attack?
  2. Focus on evaluating factors associate with future risk of  AE and Mortality. Table 1 should be the general data of this cohort. The characteristics should include Age, Gender, Lung function, GOLD Group, Comorbidity, Smoking status and the major outcomes that you want to address in this paper--AE and mortality.
  3. Table 3A and 3B should be reordered to Figure 2A and 2B (AE risk)
  4. Table 1A and 1B should be reordered to Figure 3A and 3B (Mortality risk)
  5. Delete previous Figure 2A and 2B (ICU admission is not your primary goal)
  6. Figure 4 is good but redundant. Suggest remove factors in the univariable analysis which are not significant in new Table  3A and 3B (Mortality)
  7. Since there are four factors associate with future risk of mortality, the discussion part should focus on these four factors (Age, PO2. NLR, GOLD stage).

  8. The association among "Age, PO2, NLR and GOLD stage" and future risk of mortality is not a new concept, I recommend the author to use this four parameters to form a new score and check its predictive power of mortality.  

Author Response

SECOND REVISION LETTER FOR REVIEWER 3

1-Our title was chosen in this way because our aim is to determine the factors affecting the severity of the disease (admission to the next hospital and intensive care unit), as well as subsequent attacks and mortality in COPD patients hospitalized with acute exacerbations.

2-In order not to take up much space in our article, the general data are specified as supplementary tables 5A and 5B in the supplemental file.

3-The correction you requested in Tables 3A and 3B could not be understood, as there are only tables in our article, and figures 2A and 2B are not.

4-The correction you requested in Tables 1A and 1B could not be understood, as there are only tables in our article, and figures 3A and 3B are not.

5-As can be understood from the title of our article, Tables 2A and 2B are left as they are, since one of our primary goals is to follow up hospitalization and intensive care admissions, which are parameters that show the severity of the disease in hospitalized patients with COPD attack.

6- You suggested a cox regression analysis in your previous revision (Since this cohort has been followed up for at least 6 months, cox-regression model could be used to evaluate the time effect. The statisitic part in the presented tables only using Mann-whitney u test and X² Chi square test, therefore we cannot understand the time effect of these parameters). Therefore, Table 4 showing the parameters affecting survival has been added to the article.

Tables 3A and 3B show admissions to hospital with an attack during follow-up, not mortality.

We think that it would be more appropriate to evaluate the significant and insignificant factors together in the tables.

7-In the conclusion part of our article, the importance of these factors in the prognosis of the disease has already been emphasized. In addition, the result of each parameter is discussed in detail in the discussion section.

8-We think that it would be more appropriate to make a new scoring system, which will be created from the parameters that are significant, according to the results obtained by longer follow-up of a larger patient population, in terms of determining its place in predicting the prognosis. We have stated this opinion in the conclusion part.

Reviewer 4 Report

After the corrections the manuscript can be reconsidered .

Author Response

You said that "After the corrections the manuscript can be reconsidered ." But according to your previous requests, our article has already been revised.

REVISION LETTER FOR REVIEWER 4

1.In our study, we decided that our observation period would be 1 year. We were getting information about the patients by phone every month. However, since some of the patients could not be reached after 6 months, we decided to have a follow-up period of 6 months.

Our article has been revised in English. Mistake of in mMRC has been corrected.

MATERIAL: The number of our patients with sleep-apnea was two. However, it was not specified because it was statistically low (2/160=0.012).

RESULTS: The results of pulmonary function tests (FEV1, FVC, FEV1/FVC) and arterial blood gases (pH, PO2, PCO2, O2 saturation) are already shown among the “laboratory data” in Table 1B.

DISCUSSION: “Some researchers stated that mortality is associated with a very low muscle ratio from low BMI, and more accurate results can be obtained by evaluating lean body weight instead of evaluating total body weight, but since only BMI was looked at in our study, no comparison was made with lean body weight. ” This has been added to the discussion as additional information in article.

CONCLUSION: First sentence “These findings suggest that advanced age, long COPD duration, low PFT values, high dyspnea scores, increased inflammatory markers, low BMI and anemia are the most effective factors in the determining prognosis of patients with COPD attack.” changed to.

 The second sentence was removed from the article.

The third sentence is left the same.

“We think that more detailed information on this matter can be obtained with new prospective studies that include longer follow-up periods (especially 1 year, 5 years in terms of mortality) and larger number of patients.” sentence added.

REFERENCES: Three new article (4., 16. and 27.references) have been added. References have been corrected according to the spelling rules of the journal.

DATA AVAILABILITY STATEMENT: Corrected as “The dataset that supports the findings of this study is available on request from the corresponding author”.

Revisions made within the article are indicated in bold red and underlined.

English gramer revisions made within the article are indicated in blue.

Round 3

Reviewer 3 Report

No more questions regarding this manuscript.